# Measuring the Tree Height of *Picea crassifolia* in Alpine Mountain Forests in Northwest China Based on UAV-LiDAR

Siwen Chen, Yanyun Nian *, Zeyu He and Minglu Che

College of Earth and Environmental Sciences, Lanzhou University, Lanzhou 730000, China;
chensw20@lzu.edu.cn (S.C.); hezy19@lzu.edu.cn (Z.H.); cheml21@lzu.edu.cn (M.C.)
* Correspondence: yynian@lzu.edu.cn

**Abstract:** Forests in alpine mountainous regions are sensitive to global climate change. Accurate measurement of tree height is essential for forest aboveground biomass estimation. Unmanned aerial vehicle light detection and ranging (UAV-LiDAR) in tree height estimation has been extensively used in forestry inventories. This study investigated the influence of varying flight heights and point cloud densities on the extraction of tree height, and four flight heights (i.e., 85, 115, 145, and 175 m) were set in three *Picea crassifolia* plots in the Qilian Mountains. After point cloud data were classified, tree height was extracted from a canopy height model (CHM) on the basis of the individual tree segmentation. Through comparison with ground measurements, the tree height estimations of different flight heights and point cloud densities were analyzed. The results indicated that (1) with a flight height of 85 m, the tree height estimation achieved the highest accuracy ($R^2 = 0.75$, RMSE = 2.65), and the lowest accuracy occurred at a height of 175 m ($R^2 = 0.65$, RMSE = 3.00). (2) The accuracy of the tree height estimation decreased as the point cloud density decreased. The accuracies of tree height estimation from low-point cloud density ($R^2 = 0.70$, RMSE = 2.75) and medium density ($R^2 = 0.69$, RMSE = 2.80) were comparable. (3) Tree height was slightly underestimated in most cases when CHM-based segmentation methods were used. Consequently, a flight height of 145 m was more applicable for maintaining tree height estimation accuracy and assuring the safety of UAVs flying in alpine mountain regions. A point cloud density of 125–185 pts/m$^2$ can guarantee tree height estimation accuracy. The results of this study could potentially improve tree height estimation and provide available UAV-LiDAR flight parameters in alpine mountainous regions in Northwest China.

**Keywords:** tree height; alpine forest; UAV-LiDAR; flight height; point cloud density

## 1. Introduction

*Picea crassifolia* (Kom., 1923) is a typical vegetation in Northwest China and is mainly distributed in the Qilian Mountains (approximately $1.3 \times 10^5$ ha), accounting for over 94.6% of the total area of *Picea crassifolia* in this region [1]. *Picea crassifolia* within the Qilian Mountains has been found mostly at altitudes of 2500–3200 m and in shady and semi-shady alpine areas with slopes of 12–24° [2]. As one of the main species of water conservation forest in the Qilian Mountains, correctly monitoring the forest structural parameters of *Picea crassifolia* is vital for the ecological protection and sustainable development of the Qilian Mountains.

Forest inventory in the mountainous regions of Northwest China is extremely challenging due to the presence of external constraints such as high altitude and inaccessibility. LiDAR is an active remote sensing method that has been utilized frequently in forests. LiDAR uses laser energy to determine the distance between the sensor and the target, providing accurate information regarding the forest structure, especially vertical structure information such as tree height. LiDAR can acquire forest structural parameters more conveniently and quickly than traditional field inventory methods [3–7]. As one of the major parameters of forest structure, tree height can help provide forest aboveground

biomass [8–12]. Increasing the precision of tree height extraction in point cloud data can significantly improve the estimation of aboveground biomass [8–11].

Terrestrial laser scanning (TLS) provides dense point cloud data with millimeter-level resolution from which forest structure indicators can be accurately extracted [13]. Although TLS is a good alternative to traditional forestry survey methods, as it is a nondestructive method, it is more difficult to use TLS in inaccessible mountainous areas, and the spatial extent of one performance acquisition by TLS is relatively small [14]. TLS is often suitable for accurately acquiring forest structure features in small areas and the reconstruction of forest 3D structures. In terms of estimating tree height, TLS tends to be underestimated [15,16]. TLS is better for the bottom of the canopy measurement and is suitable for extracting parameters such as DBH (diameter at breast height) [17,18], but detecting the canopy top is problematic, as the accuracy of tree height extraction is lower compared to ALS or UAV-LiDAR [7].

Compared to TLS, airborne laser scanning (ALS) has a clear advantage in estimating tree height with a much larger measurement range. However, ALS seems to be more suitable for areas with flat terrain. The point cloud density obtained with ALS is usually between 1 and 25 pts/m$^2$, and it is more expensive to use [19,20]. In recent years, with the increase in UAV loads and the availability of lightweight LiDAR sensors, some studies have integrated LiDAR sensors with UAV platforms [21,22]. Therefore, it is a wise choice to use UAV-LiDAR in mountainous areas, as it is cheaper and more flexible in flight. UAV-LiDAR can change flight parameters to better capture forest structure information for sample plots with different characteristics [23,24].

Nevertheless, there are still difficulties in using UAV-LiDAR for forest inventory in alpine mountains with relatively large terrain changes and dense forests. Using UAV-LiDAR to acquire forest parameters in mountain forests is a simple and quick approach; however, there are still limitations such as easy loss of signal and the possibility of hitting obstacles when flying the UAV. With the increase in flight height, the safety of the UAV is improved, while the point cloud density is lowered. Finally, this will cause the accuracy of the tree height estimation to decrease. Therefore, proper flight height and point cloud density are necessary to strike a balance between flight safety and data quality. In addition, flying UAVs in mountainous areas is also affected by the terrain, tree species, tree height, and performance of the LIDAR itself. Some studies have been conducted to investigate the relationship between the accuracy of tree height estimation and different point cloud densities [25] or sensors [26]. Furthermore, the majority of current studies on tree height extraction have been undertaken in areas with lower altitude and plain terrain [27–29], with fewer studies on tree height in alpine mountainous regions. As a result, for operational mountainous forest management, dependable, cost-effective methods are required for practical tree attributes [30].

In order to determine the UAV-LiDAR flight parameters suitable for collecting the tree heights of *Picea crassifolia* forests in the alpine mountains of Northwest China, we collected data at different flight heights and point cloud densities. These data were used to investigate (1) the optimal flight height for tree height estimation and (2) the appropriate point cloud density for tree height estimation in alpine mountainous areas in Northwest China.

## 2. Materials and Methods

### 2.1. Study Area

The tree species studied was a *Picea crassifolia* forest in the eastern part of the Qilian Mountains in Northwest China, which is a tall coniferous forest, using sample plots located above 2800 m (36°41′ N, 102°50′–102°50′30″ E) (Figure 1b). The growth of mature *Picea crassifolia* forests tends to be slow [31], and the selected sample plots are all mature forests. This area has a special geographic location and is a transitional intersection of three geo-ecoregions (including the eastern monsoon, northwestern dry, and Qinghai–Tibet plateau geo-ecoregions) (Figure 1a) [32], and the total forest area is approximately 25,000 ha, which

is the largest distribution area of pure *Picea crassifolia* forest in the Qilian Mountains at present, according to Google Earth [33].

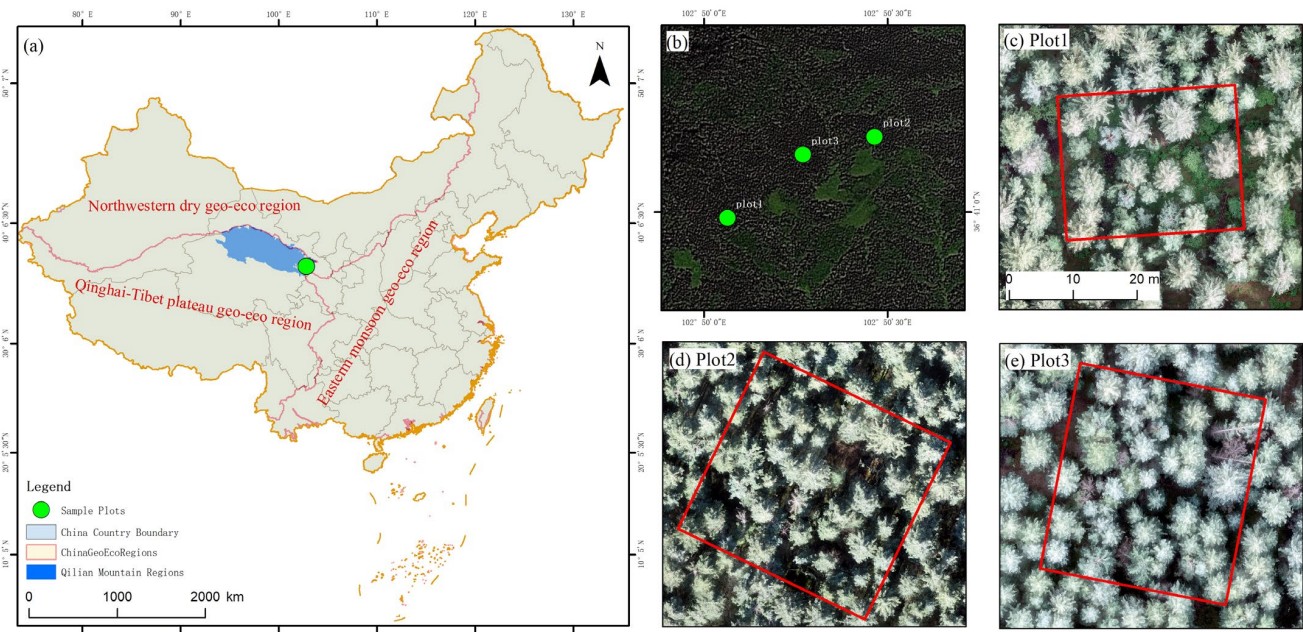

**Figure 1.** The specific site of the sample plots in China and the Qilian Mountains (**a**); the locations of three plots in a Google Earth satellite image (**b**); orthophotos of the three plots (**c**–**e**).

### 2.2. Datasets and Methods

#### 2.2.1. UAV-LiDAR Data

The locations of the three plots are shown in Figure 1b, and the orthophotos of these plots are shown in Figure 1c–e. Figure 2a shows an overview of Plot 2. LiDAR point cloud data within the sample plots were collected using the RIEGL miniVUX-1 UAV (minimum measurement level of airborne laser scanner) (Figure 2b), which was carried out using a DJI M600 Pro on 17 June and 10 October 2021.

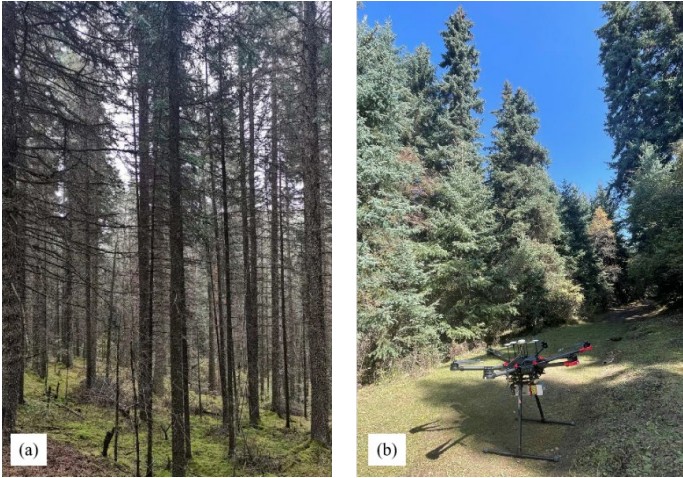

**Figure 2.** An overview of *Picea crassifolia* Plot 2 (**a**); the condition of the UAV-LiDAR (**b**).

RTK-GPS measurement data were obtained from the UniStrong GPS base station. The specific parameters of the LIDAR are shown in Table 1. Four different heights were flown above the ground surface at all three plots, taking into account the complex terrain in the mountainous area; 85 m was the minimum safe height that could be flown, and the speed

of each flight was 4 m/s. In this study, four flight heights (i.e., 85, 115, 145, and 175 m) (Table 2) were set in three *Picea crassifolia* plots, and UAV-LiDAR data were categorized with three point cloud densities. The forest cover calculated for each sample plot at four flight heights was calculated using Fusion software (USDA Forest Service, Pacific Northwest Research Station).

**Table 1.** The parameters of the UAV-LiDAR.

| Metrics | Parameters |
|---|---|
| Maximum field of view | 360° self-adjusting |
| Maximum pulse emission frequency | 100 kHz |
| Distance measuring accuracy | 10 mm |
| Ranging repeatability accuracy | 15 mm |
| Maximum distance range | 250 m |
| Laser levels | Level 1 human eye safety laser |
| Size | 243 mm × 99 mm × 85 mm |
| Weight | 1.85 kg |
| Power consumption | 16 W |
| Inertial navigation unit | IMU/GNSS |
| Plane positioning accuracy | 0.05 m |
| Elevation positioning accuracy | 0.1 m |

**Table 2.** Flight height, point cloud density of the UAV-LiDAR, and the sample plot cover obtained.

| Plot | Flight Height (m) | Average Density of the Point Cloud (pts/m$^2$) | Cover (%) |
|---|---|---|---|
| 1 | 85 | 555 | 81.25 |
| | 115 | 435 | 79.87 |
| | 145 | 318 | 78.05 |
| | 175 | 125 | 76.89 |
| 2 | 85 | 388 | 88.16 |
| | 115 | 254 | 85.81 |
| | 145 | 185 | 84.71 |
| | 175 | 134 | 83.88 |
| 3 | 85 | 351 | 85.19 |
| | 115 | 306 | 84.02 |
| | 145 | 184 | 82.35 |
| | 175 | 144 | 81.76 |

### 2.2.2. DOM Data

As auxiliary data for individual tree segmentation, the digital orthophoto model (DOM) can help to identify the accuracy of individual tree extraction. The DOM data were acquired on 17 June and 10 October 2021, using a DJI Phantom 4 RTK as a tool for obtaining orthophotos. The flight parameters were consistent at each plot (85 m and 5 m/s). Image orthomosaic was used with Pix4D Mapper software.

### 2.2.3. Ground Measurement Data

The measured data were obtained in July 2018 including tree height (TH) and diameter at breast height (DBH) (Table 3). To subsequently match the estimated data, the coordinates of the sample plots and the relative $x$ and $y$ coordinate positions of each tree within the sample plots were also measured. The TH was measured using a laser rangefinder, and in each sample plot, only trees with a DBH > 5 cm were measured with tape. Finally, every tree that was measured was recorded in the sample plots with its relative coordinates.

**Table 3.** Statistical data of each sample plot.

| Plot | Area (m$^2$) | Slope (°) | Tree Number | DBH (cm) | | TH (m) | |
|---|---|---|---|---|---|---|---|
| | | | | Mean | SD | Mean | SD |
| 1 | 28 × 25 | 14 | 43 | 26.50 | 14.36 | 18.76 | 7.54 |
| 2 | 32 × 30 | 17 | 73 | 21.85 | 10.67 | 15.41 | 5.47 |
| 3 | 32 × 30 | 19 | 56 | 24.42 | 15.14 | 15.91 | 8.05 |

DBH = diameter at breast height; TH = tree height.

### 2.2.4. Data Preprocessing

First, to reduce the gross error in the raw point cloud data, the base station data and the raw inertial measurement unit (IMU) data were input into POSPacUAV software to obtain smoothed best estimate of trajectory (SBET) data as postprocessed POS data. Second, raw point cloud and postprocessed POS data were added into RiPROCESS software. The collected data were projected onto the plane coordinates for automatic detection, and then the LiDAR point cloud was automatically aligned and adjusted according to the precision trajectory. The connection surface was fully extracted automatically, and the point cloud alignment parameters were adjusted including the system calibration information. The data were stitched together to export the point cloud data in a .las file. Third, the point cloud data were denoised using LiDAR360 4.0 software (Green Valley, Beijing, China). Finally, the LiDAR data were classified into ground points and nonground points using an improved progressive TIN densification filtering algorithm [34].

### 2.2.5. Individual Tree Segmentation

The classified point cloud was interpolated by a TIN triangulation network to obtain a digital surface model (DSM) and a digital elevation model (DEM), and a CHM was obtained by subtracting the DEM from the DSM. Segmenting CHMs into individual trees was performed using the watershed segmentation algorithm [35]. The precision rates (p) and recall rates (r) were applied to determine the accuracies of individual tree segmentation according to Equations (1) and (2), where p represents the proportion of correct segments to the extracted data; r represents the proportion of correct segments to the actual data; TP represents the number of correct segments; FN represents the missed segments; FP represents oversegmentation. A flow chart of the method used in this study is shown in Figure 3.

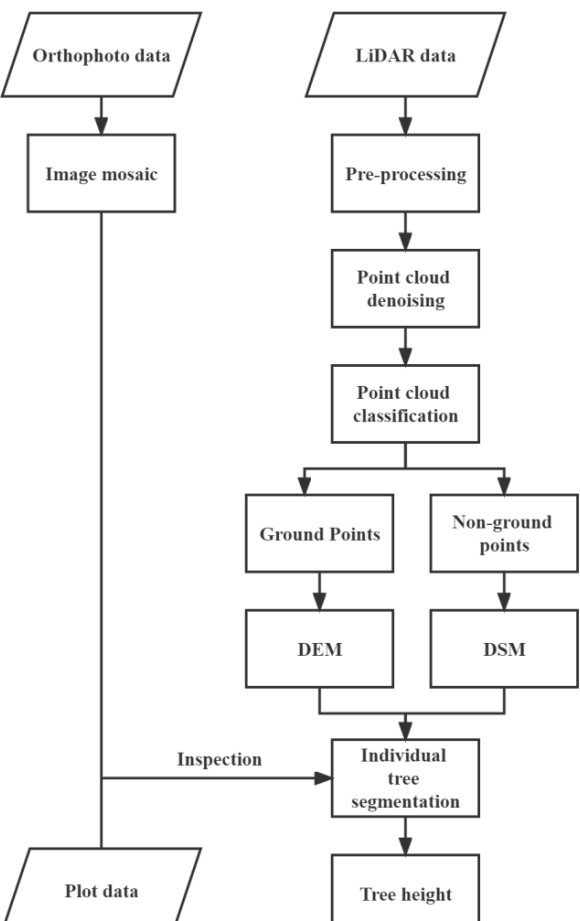

**Figure 3.** Flow chart for processing the cloud point data and estimation of tree height.

$$p = \frac{TP}{TP + FP} \tag{1}$$

$$r = \frac{TP}{TP + FN} \tag{2}$$

2.2.6. Accuracy Assessment

Field measurement data were treated as a reference for tree height. The evaluation and validation of the estimated tree height and the reference tree height using the determination coefficient ($R^2$) represents the reliability of the estimated tree height. To express the deviation of the estimated value from the actual value, we used the root mean square error (RMSE); using bias (B) to determine how much the estimated tree height was overestimated or underestimated relative to the reference value. The equations for the RMSE, $R^2$, and bias values are as follows:

$$RMSE = \sqrt{\frac{\sum_{i=1}^{n} (P_i - O_i)^2}{n}} \tag{3}$$

$$R^2 = 1 - \sum_{i=1}^{n} \frac{(P_i - O_i)^2}{(O_i - \overline{O_i})^2} \tag{4}$$

$$B = \frac{\sum_{i=1}^{n} (P_i - O_i)}{n} \tag{5}$$

In Equations (3) and (4), $P_i$ represents the estimated tree height, $O_i$ represents the reference tree height, $\overline{O_i}$ represents the mean reference tree height, and n represents the number of all correctly identified trees.

## 3. Results

### 3.1. Tree Height Estimation with Different Flight Heights

As an example, the results obtained with a flight height of 85 m in Plot 1 showing the individual tree segmentation are depicted in Figure 4c, where the green dots represent correctly segmented trees, and the red dots represent oversegmented trees. For some trees with large canopies, there was an obvious oversegmentation phenomenon. The recall rates (r) and precision rates (p) obtained from different flight heights are shown in Table 4. The flight height of 85 m had the largest number of correct trees with the largest recall rates. The 175 m height had the smallest recall rates, indicating that the number of correctly detected individual trees gradually decreased with the increase in flight height. The 145 m flight height had the largest precision rates, indicating that the phenomenon of oversegmentation decreased as the flight height increased.

Different flight heights to measured tree heights were fitted in the plots (Figure 5). The regression analysis of the LiDAR data and measured data showed that the $R^2$ value was the highest and the RMSE was the lowest at the 85 m flight height, which were 0.75 and 2.65, respectively (Figure 5a); the $R^2$ value was the lowest and the RMSE was the highest at the 175 m flight height, which were 0.65 and 3.00, respectively (Figure 5c). At the flight heights of 115 and 145 m, the fit was better, and the RMSE was smaller at 145 m.

**Table 4.** Individual tree segmentation validation.

| Flight Height (m) | r (%) | p (%) |
|:---:|:---:|:---:|
| 85 | 90.7 | 69.0 |
| 115 | 88.4 | 64.4 |
| 145 | 81.4 | 71.4 |
| 175 | 76.7 | 68.0 |

r (%) = recall rates; p (%) = precision rates.

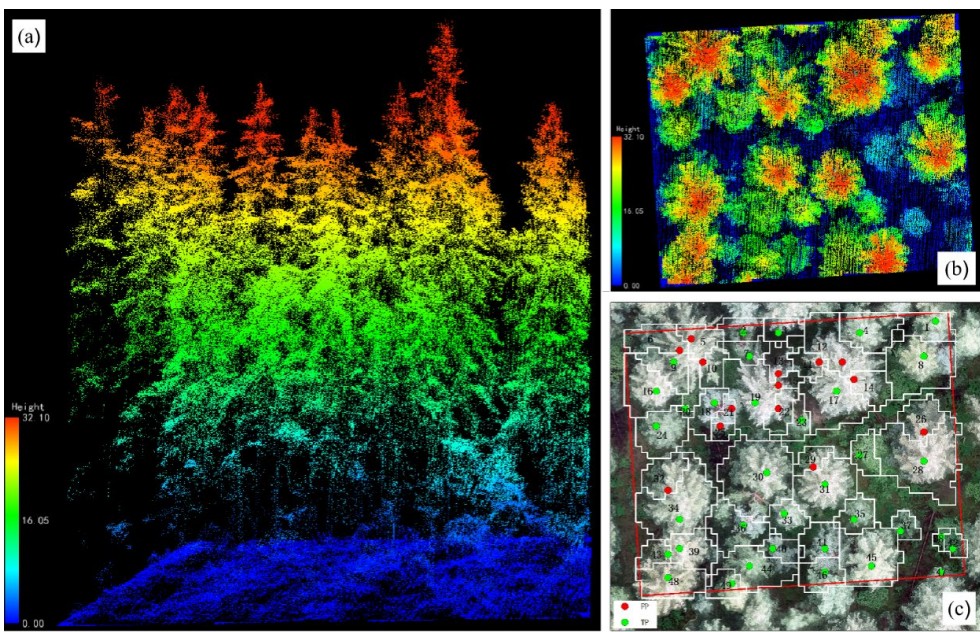

**Figure 4.** The 85 m flight height of Plot 1 is shown as an example: a front view of Plot 1 (**a**); top view of Plot 1 (**b**); results of the individual tree segmentation of Plot 1 (**c**).

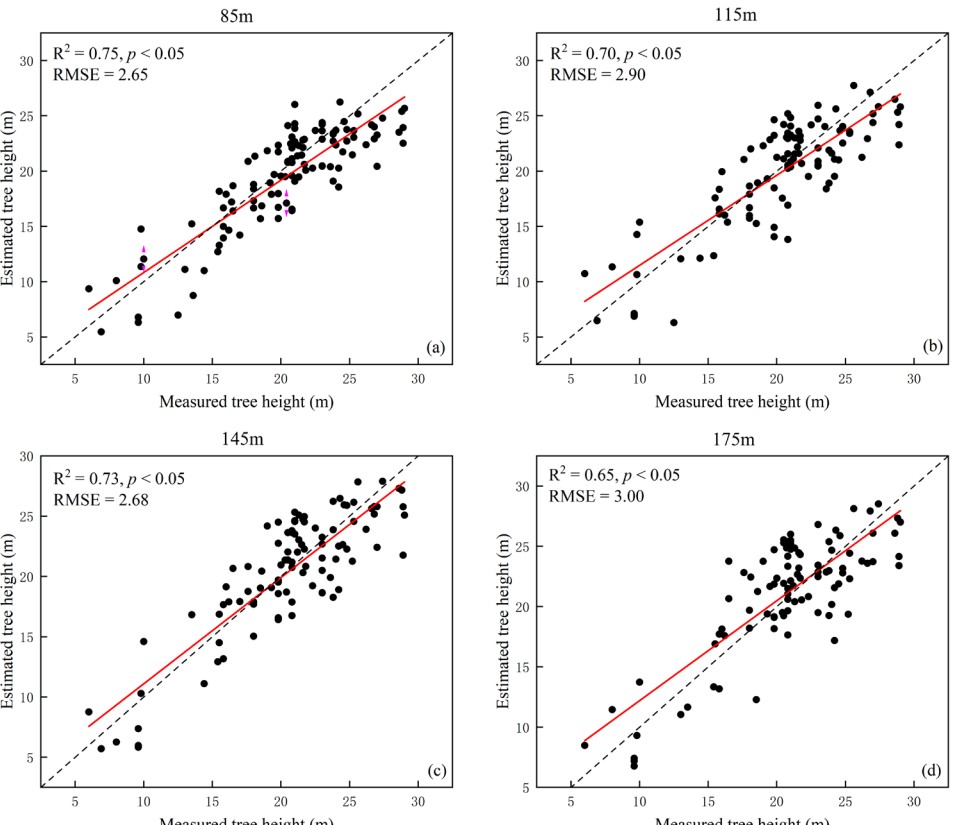

**Figure 5.** Scatter plots between estimated tree height and measured data at different flight heights. Subplots (**a**–**d**) represent flight heights of 85, 115, 145, and 175 m. The fitted line is shown by the red, solid line, while the 1:1 line is represented by the dashed line.

In addition, Figure 6 shows the estimated tree heights obtained from the four flight heights. The mean differences between 85, 115, and 145 m were less than zero (Figure 6, Table 5), which can be explained by the fact that the estimated data were smaller than

the measured data in the field, and the estimated tree heights obtained from 85 m were obviously smaller than the measured data. With the increase in flight height, the estimated tree height gradually increased, but at 145 m, the increase was not obvious. However, the mean difference at 175 m was greater than zero. In terms of bias, the 145 m flight height had the least bias. As the flight height increased, the tree height changed from underestimation to overestimation.

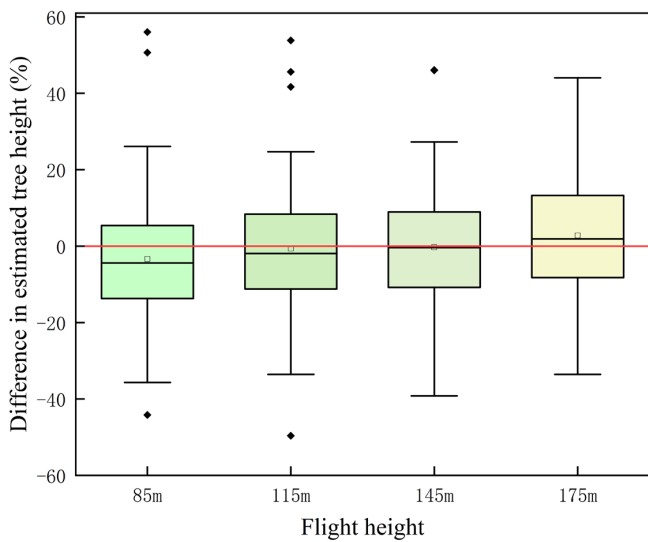

**Figure 6.** Box plots of the percentage difference in tree heights extracted at different flight heights relative to the measured tree height. The "square" symbols means the average value, "diamond" symbols means abnormal value.

**Table 5.** Accuracy assessment of the tree height extraction at different flight heights.

| Flight Height (m) | $R^2$ | RMSE | B | Mean Difference (%) |
|---|---|---|---|---|
| 85 | 0.75 | 2.65 | −0.85 | −3.38 |
| 115 | 0.70 | 2.90 | −0.41 | −0.64 |
| 145 | 0.73 | 2.68 | −0.17 | −0.30 |
| 175 | 0.65 | 3.00 | 0.37 | 2.78 |

### 3.2. Tree Height Estimation with Different Point Cloud Densities

Plot 1, with the highest point cloud density at a flight height of 85 m, was used as the reference. The point clouds obtained from different flight heights were classified into high-density point clouds (70%–100%), medium-density point clouds (40%–70%), and low-density point clouds (10%–40%) (Table 6). Figure 7 shows the morphology of a tree in Plot 1 with high-density, medium-density, and low-density point clouds. The high-density point cloud had the most complete detection of the overall morphology of the tree, and the DBH of the tree could also be detected.

**Table 6.** Different densities of point clouds and percentages of point clouds.

| Density of Point Cloud | Plot and Flight Height (m) | Percentage of Point Cloud (%) |
|---|---|---|
| High density | Plot 1 at 85 m | 100 |
| | Plot 1 at 115 m | 78 |
| | Plot 2 at 85 m | 70 |
| Medium density | Plot 1 at 145 m | 57 |
| | Plot 2 at 145 m | 46 |
| | Plot 3 at 85 m | 63 |
| | Plot 3 at 115 m | 55 |
| Low density | Plot 1 at 175 m | 23 |
| | Plot 2 at 145 m | 33 |
| | Plot 2 at 175 m | 24 |
| | Plot 3 at 145 m | 33 |
| | Plot 3 at 175 m | 26 |

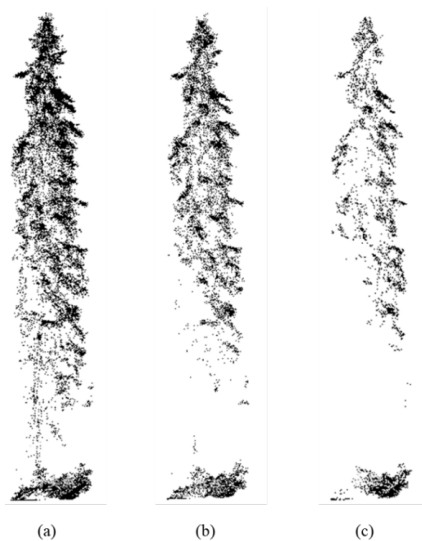

**Figure 7.** (**a**–**c**) Morphology of a tree in Plot 1 in the high-density point cloud, medium-density point cloud, and low-density point cloud, respectively.

The fit of individual tree heights extracted from the LiDAR data at different densities of point cloud to the measured tree heights is shown in Figure 8. The extracted tree height from the high-density point cloud was the best fit to the measured data ($R^2 = 0.78$). The extracted tree height from the low-density point cloud was the worst fit to the measured data ($R^2 = 0.69$). However, the results were different in terms of root mean square error, with the high-density point cloud having the highest error (RMSE = 2.88), the medium-density point cloud having the lowest error (RMSE = 2.75), and the low-density point cloud having an error somewhere in between. The accuracies of tree height estimation for medium-point cloud density ($R^2 = 0.70$, RMSE = 2.75) and low-point cloud density ($R^2 = 0.69$, RMSE = 2.80) were very similar.

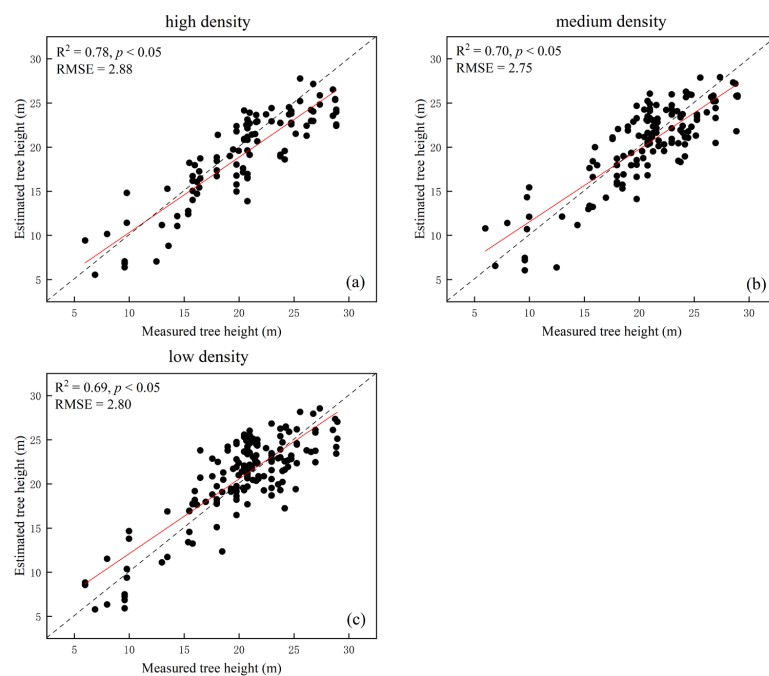

**Figure 8.** Scatter plots between the estimated tree height and the measured data at different densities of point cloud. Subplots (**a**–**c**) represent the high-density point cloud, medium-density point cloud, and low-density point cloud. The fitted line is shown by the red, solid line, while the 1:1 line is represented by the dashed line.

The bias of the high-, medium-, and low-density point clouds were −1.23, −0.45, and 0.43, respectively. The mean differences were −8.83%, −3.78%, and 0.62% (Figure 9, Table 7). We found that low-density point clouds had the smallest bias and mean difference, high-density and medium-density point clouds underestimated tree height, and low-density point clouds overestimated tree height.

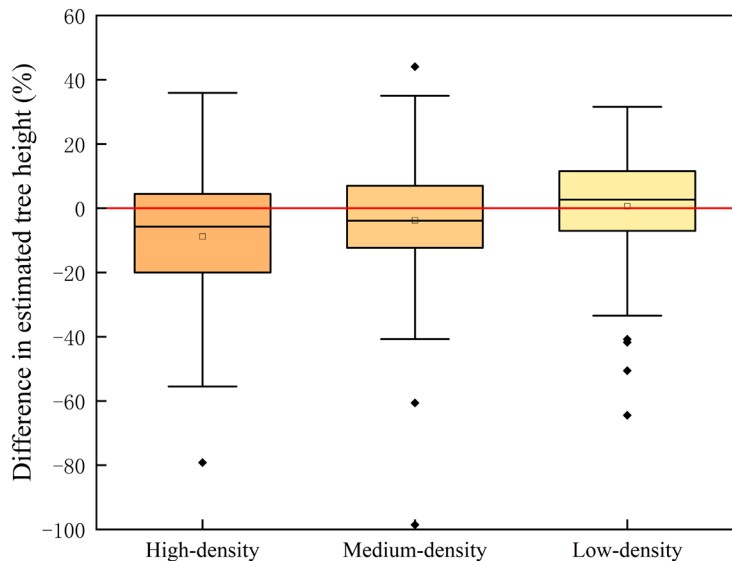

**Figure 9.** Box plots of the measured field data and the different point cloud densities of the LiDAR data. The "square" symbols means the average value, "diamond" symbols means abnormal value.

**Table 7.** Accuracy assessment of tree height extraction at different point cloud densities.

| Point Cloud Density | R2 | RMSE | B | Mean Difference (%) |
| --- | --- | --- | --- | --- |
| High density | 0.78 | 2.88 | −1.23 | −8.83 |
| Medium density | 0.70 | 2.75 | −0.45 | −3.78 |
| Low density | 0.69 | 2.80 | 0.43 | 0.62 |

## 4. Discussion

The flight height correlated with the accuracy of individual tree segmentation. As the flight height increased, the number of correct individual tree segmentations decreased, and the recall rates (r) decreased from 90.7% to 76.7%, indicating that the undersegmentation phenomenon increased. This was due to the decrease in point cloud density caused by the increase in flight height. However, too dense point clouds can also cause the oversegmentation phenomenon. Among them, the 145 m flight height had the best performance in oversegmentation.

With the flight height increase, the oversegmentation phenomenon can be reduced to a certain extent. The precision rates (p) of the 145 m flight height was significantly smaller than that of 115 m, because the CHM identified neighboring and similar raster values as individual tree vertices at higher point cloud densities. The estimation accuracy of tree height was related to individual tree segmentation. Wu et al. [8] found that improving the accuracy of individual tree segmentation can advance the calculation of forest biomass. In general, the lower the flight height, the more accurate the individual tree height [26]. From a flight height of 145 m, the estimated tree height fit better than the 115 m height, probably because the number of individual trees extracted at the two flight heights was approximately the same. At a flight height of 115 m, individual trees were extracted with more severe oversegmentation (Table 4), and the estimated tree height was more different than the measured tree height (Figure 6). In terms of bias and mean difference (Table 5), 145 m was the most suitable flight height and had a certain accuracy in fitting with the measured data ($R^2$ = 0.73).

Tree height estimation at different flight heights (i.e., 85, 115, and 145 m) were lower than the measured tree height (Figure 6, Table 5). Various studies have found that individual tree segmentation-derived tree heights based on the CHM were significantly lower than those obtained from point cloud segmentation, because the interpolating process of point clouds into the CHM led to an overall reduction in tree height [36,37]. The Gaussian smoothing window was used in CHM-based individual tree segmentation, which took the average of the surrounding raster values to determine the maximum value of the top point of an individual tree, and this method caused the estimated tree height to be lower than the actual value [38–40]. In other experiments that used UAV-LiDAR to obtain tree height, the phenomenon of tree height underestimation existed. Krause et al. [41] obtained tree heights semi-automatically based on drones and found that measured tree heights were often overestimated, while drone measurements of tree heights were often underestimated. With the increase in flight height, the underestimation phenomenon was gradual and not obvious, which was probably because the acquired ground point cloud was becoming sparser, and the inaccuracy of the ground point separation caused the acquired tree height to become larger and the bias from the actual value to become smaller. The errors caused by ground points may be more obvious, especially in areas with large mountain slopes.

Various studies have found that different point cloud densities affect the estimation accuracy of tree height; the higher the point cloud density, the better the estimated tree height fits the measured tree height [25,26]. The medium-density point cloud extracted tree height with higher accuracy than the low-density point cloud, but the accuracy did not increase much, which indicates that the point cloud density at 125–185 pts/m$^2$ extracted tree height accurately. Therefore, it was not necessary to raise the point cloud density to 254–351 pts/m$^2$. Peng et al. [25] found that when the point cloud density increased to 17 pts/m$^2$, the estimation accuracy of the tree height did not improve significantly. As the point cloud density decreased, the bias tree height relative to the measured value as well as the mean difference also decreased, and some studies have similar results [42].

We found that the extracted accuracy of tree heights of *Picea crassifolia* in alpine mountainous forests was inferior to that in several existing studies [7,28,43]. Perhaps the slope of the mountainous terrain caused some uncertainty in distinguishing ground points and nonground points and then resulted in the lower extracted accuracy of tree height. In addition, the actual measurement data were obtained from rolling ground, and the measurement of tree height was affected by the slope of the mountain. Altitude and slope are the most important influencing factors when setting the flight heights of UAVs. Although we only set-up three *Picea crassifolia* sample plots in the eastern Qilian Mountains, these plots were typically representative. Our results may be applicable to the survey of *Picea crassifolia* mature forests with a certain slope. In future studies, we will add the effect of slope on the extraction of forest structure parameters in mountainous areas in Northwest China. In addition to the use of UAV-LiDAR, there is currently research on the use of the more cost-effective unpiloted digital aerial photogrammetry (UAS DAP) to predict forest attributes [44], and we can also try to use UAS DAP for forest monitoring in alpine mountain areas in the future.

## 5. Conclusions

UAV-LIDAR was utilized to acquire point cloud data in alpine mountainous areas. We found that a flight height of 145 m was more applicable to maintain tree height estimation accuracy and assuring the safety of UAVs flying in alpine mountain regions. A point cloud density of 125–185 pts/m$^2$ can guarantee tree height estimation accuracy. These results can serve as a good reference for alpine forest management. In retrospect, our study employing UAV-LiDAR might contribute to a more automated forest inventory, which could help to reduce the amount of field activities performed in the future.

In general, the estimation of tree height only requires LiDAR to detect the point cloud at the top of the tree, and the accuracy difference between different flight heights was not significant. If the ground point could be separated more accurately, the tree height would

be extracted closer to the measured value. This study may improve the flight efficiency of UAV-LiDAR in alpine mountain forests in Northwest China. By selecting the appropriate flying height and point cloud density, considerable cost efficiency and time savings may be realized, benefiting forestry employees.

**Author Contributions:** Conceptualization, Y.N., S.C. and Z.H.; methodology, S.C. and Z.H.; software, S.C.; validation, S.C., Z.H. and M.C.; formal analysis, S.C.; investigation, Z.H. and M.C.; data curation, S.C., Z.H. and M.C.; writing—original draft preparation, S.C.; writing—review and editing, Y.N.; project administration, Y.N.; funding acquisition, Y.N. All authors have read and agreed to the published version of the manuscript.

**Funding:** This research was funded by the Second Tibetan Plateau Scientific Expedition and Research Program (STEP) (grant no.2019QZKK0301) and National Key R&D Program of China (grant no.2019YFC0507401).

**Institutional Review Board Statement:** Ethical review and approval were waived for this study, since the study did not involve humans or animals.

**Informed Consent Statement:** Not applicable.

**Data Availability Statement:** The data presented in this study are available on request from the corresponding author. The data are not publicly available due to confidentiality of the data.

**Conflicts of Interest:** The authors declare no conflict of interest.

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
