# Peer review of "Measuring the Tree Height of Picea crassifolia in Alpine Mountain Forests in Northwest China Based on UAV-LiDAR"

_forests, doi:10.3390/f13081163_

Round 1

Reviewer 1 Report

The paper addresses an exciting theme and an alternative for forest inventory in mountainous regions. 

Nonetheless, the use of LiDAR to estimate tree attributes (mainly tree height) has become relatively common in the last years. The strong sides of ALS and TLS have been extensively discussed in the literature. Then, UAV-LiDAR emerged as an alternative tool due to its flexibility and data quality. However, the authors presented a superficial introduction to this thematic and abruptly proposed the evaluation of different point densities and flight heights. 

Although the trees of Picea crassifolia have grown in mountainous regions, it was expected that LiDAR data (collected three years later) presented higher tree heights than field data. In addition, using the coefficient of determination () does not look the proper way to analyze the effect of point density and flight height against field data. Please take some time to read about the comparison of means.

Rewrite the introduction, improve the methodology description and design a proper discussion, evidencing the results and applications.

Proofreading or receiving some help from a fluent researcher is suggested (Lines . 

Reviewer 2 Report

Journal: Forests (ISSN 1999-4907)

Manuscript ID: forests- 1723371

Title: Measuring the Tree Height of Picea crassifolia in Alpine Mountain Forests in Northwest China based on UAV-LiDAR

Overall  Comments and Suggestions for Authors

Dear author,

Regarding the tree height measurement using UAV-LiDAR approach especially in alpine mountain forests for Picea crassifolia, this manuscript would be interesting to the relevant researchers who deals with similar issues for field data collection such as forest biometrician, growth and yield modeler, and forest ecologist and silviculturist. I was satisfied with the overall structure and logic flows described in the manuscript. I could not find out any critical drawbacks in any sections, so I provided my comments to improve the manuscript and prepare some potential questions which might bring from other reviewers and readers.

I hope that this manuscript can be improved based on peer-review’s comments. My specific comments were offered in detail as follows.

Kind regards,

Reviewer

Point 1.

Those heights are different by plots. Are those different height (m) valid to evaluate the prediction accuracy? I rather consider that this point is one of the weak points in this manuscript.

Is the number of plots enough to make a conclusion?

Point 2.

Isn’t there any stand characteristics that would affect the accuracy of UAV data collection?

Point 3.

Would authors consider that this result can extend to the other species as well as Picea crassifolia? Or should it be only for Picea crassifolia?

Point 4.

Although authors mentioned and compared the results with other previous studies in discussion, I would like to see the author’s recommendation as a result regarding the UAV height and point densities if available.

Point 5.

Still, the number of plots and the different set of height in UAV measurements can be a controversial issue for further process. I consider it can be acceptable depending on the conditions and a degree of necessity. If available, I suggest authors provide some discussions more to support the experimental design and the number of samplings.

Minor comments.

I recommend authors not to use “and” as like adverb in the beginning of the sentences. It is a coordinate conjunction. It may not be a desirable way to express in a scientific journal article. Alternative, “Also,” or “In addition, “ can be available if needed.

I evaluated that Moderate English changes required.

Add the continuous line numbering when you resubmit the second version of the manuscript.

Abbreviations must be clarified independently in the Tables and Figures. For example, in Table 3, DBH and TH should be explained in the caption.

Reviewer 3 Report

The article aims to determine tree height of Picea crassifolia using LiDAR point count data and to evaluate the tree height results based on UAV flight height. There are several papers out there estimating tree height using LiDAR data and determining the best flight height is useful. The study is simple and using LiDAR point cloud data to estimate vegetation matrices such as above ground biomass, canopy height models, and vertical gap index could be useful.

Here are a few examples using vegetation matrices

Jung, J., Pekin, B.K. and Pijanowski B.C. 2013. Mapping open space in an old-growth, secondary-Growth, and Selectively-Logged Tropical Rainforest Using Discrete Return LIDAR. IEEE Journal of Selected Topics in Applied Earth Observations and Remote Sensing, 6:2453-2461.      

Lamping JE, Zald HSJ, Madurapperuma BD, Graham J. Comparison of Low-Cost Commercial Unpiloted Digital Aerial Photogrammetry to Airborne Laser Scanning across Multiple Forest Types in California, USA. Remote Sensing. 2021; 13(21):4292. https://doi.org/10.3390/rs13214292

In addition, authors can give some plot statistics such as % cover, minimum height, maximum height etc., which can simply derive using FUSION open source software.

Here is a tutorial if authors wish to do some plot statistics

http://gsp.humboldt.edu/OLM/Courses/GSP_326/lidar_introductory_lab/instructions.html

I did not see the line numbers in the manuscript and therefore I added comments to your PDF document.

Round 2

Reviewer 1 Report

First, I would like to acknowledge the authors for all improvements throughout the text. There are now only a few details before publication:

Line 31 – Insert the author when you mention the species for the first time;

Line 32 – Use hectares (ha) or kilometers (km²) inset of hm²;

Line 60 – Define DBH and any other variable when first mentioning it;

Line 99 – How slow? Can you provide any number for reference?

Table 2 – Please check its formatting. The number on the plots row seems scrambled;

Line 204 – Please insert space after the number and before the unit (in this case and thorough the text);

Table 3 – Use Tree Number inset of Tree numbers;

Line 251 – Determination coefficient (not the coefficient of determination coefficient);

Line 273 and 275, and Equation 5 – Always refer to Bias with B in uppercase;

Finally, please check the Tables and Figures formatting. Ensure the headings are in the same format (check the Authors' Recommendations). 

Reviewer 3 Report

The revised manuscript looks good and authors addressed most comments. As suggested authors did a few analysis of calculaying plot ststistics such as % plot cover using FUTION. Glad to know that TIN is an intermediate product of LiDAR360 software and include such information which is useful for readers.
